# Radiotherapy in Glioblastoma Multiforme: Evolution, Limitations, and Molecularly Guided Future

**DOI:** 10.3390/biomedicines13092136

**Published:** 2025-09-01

**Authors:** Castalia Fernández, Raquel Ciérvide, Ana Díaz, Isabel Garrido, Felipe Couñago

**Affiliations:** 1Department of Radiation Oncology, Hospital Universitario San Francisco de Asís, GenesisCare, 28002 Madrid, Spain; isabel.garrido@genesiscare.es (I.G.); felipe.counago@genesiscare.es (F.C.); 2Department of Radiation Oncology, Hospital Universitario Vithas La Milagrosa, GenesisCare, 28010 Madrid, Spain; 3Department of Radiation Oncology, Hospital Universitario HM Sanchinarro, HM Hospitales, 28050 Madrid, Spain; rciervide@hmhospitales.com; 4Radiation Therapy Department, Quironsalud Madrid University Hospital, Diego de Velázquez 1, Pozuelo de Alarcón, 28223 Madrid, Spain; adiazg@quironsalud.es; 5Department of Medicine, Health and Sports, European University of Madrid, Villaviciosa de Odón, 28670 Madrid, Spain

**Keywords:** glioblastoma multiforme, radiotherapy, temozolomide, MGMT methylation, proton therapy, tumor treating fields, targeted therapies, immunotherapies, personalized medicine, advanced imaging

## Abstract

Glioblastoma multiforme (GBM), the most aggressive primary brain tumor in adults, has a poor prognosis due to rapid recurrence and treatment resistance. This review examines the evolution of radiotherapy (RT) for GBM management, from whole-brain RT to modern techniques like intensity-modulated RT (IMRT) and volumetric modulated arc therapy (VMAT), guided by 2023 European Society for Radiotherapy and Oncology (ESTRO)-European Association of Neuro-Oncology (EANO) and 2025 American Society for Radiation Oncology (ASTRO) recommendations. The standard Stupp protocol (60 Gy/30 fractions with temozolomide [TMZ]) improves overall survival (OS) to 14.6 months, with greater benefits in O6-methylguanine-DNA methyltransferase (MGMT)-methylated tumors (21.7 months). Tumor Treating Fields (TTFields) extend median overall survival (mOS) to 31.6 months in MGMT-methylated patients and 20.9 months overall in supratentorial GBM (EF-14 trial). However, 80–90% of recurrences occur within 2 cm of the irradiated field due to tumor infiltration and radioresistance driven by epidermal growth factor receptor (EGFR) amplification, phosphatase and tensin homolog (PTEN) mutations, cyclin-dependent kinase inhibitor 2A/B (CDKN2A/B) deletions, tumor hypoxia, and tumor stem cells. Pseudoprogression, distinguished using Response Assessment in Neuro-Oncology (RANO) criteria and positron emission tomography (PET), complicates response evaluation. Targeted therapies (e.g., bevacizumab; PARP inhibitors) and immunotherapies (e.g., pembrolizumab; oncolytic viruses), alongside advanced imaging (multiparametric magnetic resonance imaging [MRI], amino acid PET), support personalized RT. Ongoing trials evaluating reirradiation, hypofractionation, stereotactic radiosurgery, neoadjuvant therapies, proton therapy (PT), boron neutron capture therapy (BNCT), and AI-driven planning aim to enhance efficacy for GBM IDH-wildtype, but phase III trials are needed to improve survival and quality of life.

## 1. Introduction

Glioblastoma multiforme (GBM), the most common malignant primary brain tumor in adults, has an incidence of 3–5 cases per 100,000 people annually, with a male predominance (1.6:1) and peak incidence at 50–60 years [1,2,3,4]. According to the 2021 World Health Organization (WHO) classification, GBM is defined as an isocitrate dehydrogenase (IDH)-wild-type (IDH-wt), grade 4 astrocytoma, diagnosed by histological features (necrosis; microvascular proliferation) or molecular markers, including telomerase reverse transcriptase (TERT) promoter mutation, epidermal growth factor receptor (EGFR) amplification, or +7/−10 chromosomal alterations [5]. Most GBMs arise de novo, though a small percentage develop from lower-grade gliomas; risk factors include prior cranial radiotherapy (RT) or hereditary syndromes like Cowden, Turcot, Lynch, Li-Fraumeni, or neurofibromatosis type I [2]. Multidisciplinary treatment combines maximal safe resection followed by RT and concurrent chemotherapy (CTx), with no defined second-line therapy, necessitating personalized approaches [2,6]. Standard RT techniques, such as intensity-modulated RT (IMRT) and volumetric modulated arc therapy (VMAT) for precise dose delivery, are widely used, while experimental proton therapy (PT) and boron neutron capture therapy (BNCT) are under investigation, as detailed in Section 5.1. Despite advances, limited progress in outcomes underscores the need for innovative strategies to improve oncologic results and quality of life, as explored in Section 2, Section 3, Section 4 and Section 5. This review examines the evolution, limitations, and molecularly guided advances in RT for GBM management.

### Search Strategy

To inform this review, we consulted PubMed, Scopus, and Google Scholar, focusing on clinical trials and guidelines (2023–2025) for GBM radiotherapy, using terms like ‘glioblastoma radiotherapy’, ‘GBM molecular biomarkers’, and ‘proton therapy GBM’. We prioritized English articles from 2000 to 2025, emphasizing clinical trials and guidelines, and excluded non-human studies.

## 2. Evolution of Radiotherapy in GBM: Foundations, Standards, and International Guidelines

GBM is characterized by rapid progression and treatment resistance, with a median overall survival (mOS) of 12–15 months and a 5-year survival rate below 10%, varying by prognostic factors such as age, functional status (Karnofsky Performance Status [KPS]), O6-methylguanine-DNA methyltransferase (MGMT) promoter methylation, tumor location, and involvement of deep structures or functional areas [6,7,8,9]. The first-line treatment for GBM involves maximal safe resection, followed by adjuvant RT and CTx to target residual microscopic disease due to the tumor’s infiltrative nature [10,11]. Preoperative neuronavigation magnetic resonance imaging (MRI) and early postoperative contrast-enhanced MRI (within 48–72 h) are recommended, with intraoperative biopsy for diagnosis confirmation if resection is not feasible [2,10]. RT should begin 3–6 weeks post-surgery to minimize recurrence risk [10,11]. The standard protocol, established by the European Organisation for Research and Treatment of Cancer (EORTC) 26981/22981 trial, combines normofractionated RT (60 Gy in 30 fractions) with concomitant temozolomide (TMZ, 75 mg/m^2^/day), followed by 6 cycles of adjuvant TMZ (150–200 mg/m^2^/day, 5/28 days), improving overall survival (OS) from 12.1 to 14.6 months, particularly in patients with MGMT promoter methylation (Table 1) [12]. This regimen is recommended for patients ≤70 years with good functional status (Karnofsky Performance Status [KPS] ≥60) [13,14]. Extending TMZ beyond 6 cycles lacks evidence [15]. Tumor Treating Fields (TTFields), which disrupt cell division with alternating electric fields, are initiated 4–7 weeks post-radiotherapy during maintenance TMZ in selected patients (supratentorial GBM; KPS ≥60), improving mOS to 31.6 months in MGMT-methylated patients and 20.9 months overall (EF-14 trial), with minimal toxicity (mainly skin irritation), and are endorsed by National Comprehensive Cancer Network (NCCN) and American Society for Radiation Oncology (ASTRO) 2025 guidelines (Table 1) [12,16,17]. Other therapeutic innovations and systemic treatment options are discussed in the therapeutic innovations section.

Hypofractionated RT (40.05 Gy/15 fractions with TMZ, 34 Gy/10 fractions, or 25 Gy/5 fractions without TMZ) is the preferred standard of care for elderly (≥70 years) or frail patients (KPS 50–70), offering comparable overall survival (OS), significantly reduced toxicity, and enhanced quality of life, as supported by clinical trials and meta-analyses (Table 2) [18,19,20,21,22,23,24]. ASTRO 2025 and European Society for Radiotherapy and Oncology (ESTRO)-European Association of Neuro-Oncology (EANO) 2023 guidelines conditionally recommend these regimens for elderly or frail patients, while those with poor KPS (≤40) or extreme frailty may receive hypofractionated RT alone, TMZ alone (if MGMT-methylated), or palliative care to prioritize quality of life [10,11,13,14,25].

RT techniques have evolved from whole-brain RT (WBRT) in the 1970s to three-dimensional conformal RT (3D-CRT) in the 1990s, and now to IMRT and VMAT, which optimize dose delivery and spare healthy tissues. Treatment planning integrates thin-slice CT and MRI (T1 with gadolinium, T2/FLAIR) performed ≤14 days before RT for precise tumor and organ-at-risk delineation. ASTRO 2025 and ESTRO-EANO 2023 [11,25] guidelines prioritize IMRT/VMAT over 3D-CRT to reduce neurological toxicity. ESTRO-EANO 2023 recommends a single-phase approach (gross tumor volume [GTV]: surgical cavity + T1 enhancement; clinical target volume [CTV]: GTV + 15 mm; edema optional only for non-hyperintense T1 tumors), while ASTRO 2025 allows a single phase (edema optional) or two phases (initial phase with T2/FLAIR; boost without edema), with CTV of 10–20 mm and planning target volume (PTV) of 2–5 mm with image-guided RT (IGRT) (Table 3) [11,12,14,25,26,27,28,29]. A randomized trial with 245 patients with grade 3–4 gliomas compared the RTOG/NRG approach (including edema with a boost) and the EORTC approach (including edema in the initial phase), finding no significant differences in grade 3–4 toxicity (36.1% vs. 32.3%) or recurrence patterns (predominantly central), though neurocognitive outcomes were not assessed [30,31]. These findings, consistent with prior studies, support the flexibility of ASTRO 2025 and ESTRO-EANO 2023 guidelines, which allow either single-phase (edema optional) or two-phase (edema in initial phase) approaches (Table 3) [11,25].

The MD Anderson Cancer Center (MDACC) protocol, excluding edema, improved OS (17 vs. 12 months) and quality of life without increasing toxicity, suggesting smaller volumes may be beneficial (Table 3) [27].

Post-treatment follow-up includes MRI at 4 weeks, then every 2–4 months, per Response Assessment in Neuro-Oncology (RANO) criteria. Pseudoprogression, challenging to distinguish from true progression, may require advanced imaging (diffusion MRI, spectroscopy MRI, or amino acid positron emission tomography (PET)/CT), correlating findings with conventional MRI. In case of progression, a personalized approach is recommended, considering tumor characteristics (size, location, and molecular profile), initial treatment response, age, KPS, symptoms, needs, and patient preferences, with multidisciplinary evaluation of options (reoperation, reirradiation, or systemic therapy) [11,25]. These advances underscore the need for tailored RT strategies to optimize GBM treatment outcomes.

## 3. Limitations of Radiotherapy in GBM

RT combined with TMZ is a cornerstone of GBM treatment but faces technical, biological, and clinical limitations that reduce efficacy and lead to high recurrence and resistance rates. These barriers, summarized in Table 4, underscore the need for innovative strategies discussed later.

Technically, 90% of GBMs recur within 2 cm of the irradiated field due to diffuse infiltration [32], despite advanced techniques like IMRT, VMAT, or proton therapy (PT). High doses cause fatigue, radiation necrosis, and cognitive deficits, linked to irradiated brain volumes receiving 20 Gy (V20Gy) and 40 Gy (V40Gy), though IMRT, VMAT, and PT reduce neurotoxicity [25].

Dosimetric constraints (<54 Gy to brainstem and optic chiasm) limit dose escalation [25]. Delays >6 weeks from surgery to RT worsen OS and PFS, though moderate delays (~6 weeks) may benefit patients with residual disease [33,34].

TTFields extend PFS by 2.7 months (6.7 vs. 4.0 months) but are limited by cost and adherence (≥18 h/day) [35].

Biologically, GBM radioresistance is driven by genetic heterogeneity, including EGFR amplification (57% per TCGA), phosphatase and tensin homolog (PTEN) mutations (40% per TCGA), cyclin-dependent kinase inhibitor 2A/B (CDKN2A/B) deletions (60% per TCGA), TERT promoter mutations (>70% per TCGA), and tumor stem cells [36,39]. Additionally, the immunosuppressive tumor microenvironment, including macrophages/microglia, and neuronal crosstalk promoting proliferation and invasion contribute to resistance.

Poly (ADP-ribose) polymerase (PARP) inhibitors such as veliparib and other radiosensitizers show promise in early-phase trials by inhibiting DNA repair, awaiting phase III confirmation [36,39]. EGFR amplification activates the phosphatidylinositol 3-kinase/protein kinase B (PI3K/AKT) pathway, conferring RT resistance, with limited benefits from inhibitors like erlotinib [41].

Tumor hypoxia, mediated by hypoxia-inducible factor 1-alpha (HIF-1α), promotes resistance [36]. Strategies like modulated electrohyperthermia (mEHT) and hypoxia-sensitizing agents enhance radiosensitivity, while high-linear energy transfer (LET, energy deposited per unit length) radiation (e.g., carbon ions; alpha particles) is promising but investigational [37]. BNCT generates alpha particles in situ via neutron capture by boron in tumor cells, differing from targeted alpha therapy (TAT) using direct alpha-emitters (e.g., 213Bi, 225Ac, and 211At) conjugated to ligands like substance P for selective tumor targeting (see Section 5.6).

The subventricular zone (SVZ), a reservoir of tumor stem cells with genetic alterations, drives recurrence; irradiating the SVZ with doses ≥56 Gy (ipsilateral) or ≥50 Gy (contralateral) does not significantly improve PFS or OS [42,43].

RT-induced lymphopenia (14% with protons vs. 39% with photons) limits immunotherapy efficacy by depleting CD4 and CD8 T cells, critical for immune checkpoint inhibitors; RT may enhance immunogenicity by inducing tumor antigen release, but the immunosuppressive tumor microenvironment, including regulatory T cells (Tregs) and myeloid-derived suppressor cells (MDSCs), restricts benefits (see Section 5.5 for further details) [36,44].

Clinically, pseudoprogression (30–40% with MGMT-methylated) complicates assessment up to 12 weeks [46,47]. Multimodal imaging, including multiparametric MRI and amino acid PET enables RT personalization, achieving a mOS of 23 months in a phase I trial without significant toxicity [48].

Biomarkers (EGFR, PTEN, TERT, and MGMT) enable stratification; only MGMT methylation improves OS in phase III trials [49]. A recent meta-analysis showed that combined targeted therapies improve PFS in newly diagnosed GBM (nGBM) but not OS in nGBM or recurrent GBM (rGBM), reflecting tumor heterogeneity and molecular resistance [50].

## 4. Molecular Determinants in Glioblastoma Multiforme

Most GBMs, classified as IDH-wt, exhibit aggressive biology and radioresistance due to genetic and epigenetic alterations that enhance DNA repair and cell survival [49]. MGMT methylation (~45%) increases sensitivity to chemoradiation (CRT), while EGFR, PTEN, and TERT contribute to resistance [12,49]. Patients with MGMT methylation, a beneficial prognostic marker, show significantly higher OS (21.7 vs. 15.3 months) and better response to reirradiation in relapse compared to unmethylated cases [12,51,52] (Table 5).

A key radioresistance mechanism in IDH-wt GBMs is activation of the phosphatidylinositol 3-kinase/protein kinase B/mammalian target of rapamycin (PI3K/AKT/mTOR) pathway, driven by EGFR amplification and PTEN loss, which promotes proliferation, inhibits apoptosis, and enhances DNA repair post-RT [40]. Inhibitors like erlotinib yield modest results (ESMO Scale for Clinical Actionability of Molecular Targets (ESCAT) IIIA) [54]. Other alterations, including phosphatidylinositol-4,5-bisphosphate 3-kinase catalytic subunit alpha (PIK3CA) mutations and cyclin-dependent kinases 4 and 6 (CDK4/6) amplification, enhance cell cycle progression, while CDKN2A/B homozygous deletion, causing loss of cyclin-dependent kinase inhibitor 2A (p16^INK4a^) and alternate reading frame protein p14 (p14^ARF^), is linked to poor prognosis [49,56]. These biomarkers lack effective targeted therapies due to signaling redundancy and pharmacokinetic limitations in the central nervous system [55]. In contrast, IDH1/2 mutations are absent in IDH-wt GBMs (defining the subtype), but in rare IDH-mutant astrocytoma grade 4 (~10% of grade 4 gliomas), they confer greater radiosensitivity and better prognosis (ESCAT I) [53]. TERT promoter mutations, present in >70% of IDH-wt GBMs, activate telomerase and may promote immune evasion, though their role as predictive markers or therapeutic targets remains unclear [3,57]. Ongoing clinical trials are exploring TERT as a potential immunotherapeutic target, particularly in combination with immune checkpoint inhibitors, to enhance RT efficacy in immunosuppressive tumor microenvironments [3]. Given the limited clinical impact of standard RT in unfavorable molecular subgroups, radiosensitization strategies are being explored to enhance its therapeutic effect [58]. MGMT promoter methylation, a beneficial epigenetic marker, cannot be directly inhibited or induced pharmacologically in clinical practice; MGMT protein inhibitors (e.g., O6-benzylguanine) aim to mimic its effect in unmethylated tumors but are limited by toxicity and are not standard [51].

Among the most promising are poly (ADP-ribose) polymerase (PARP) inhibitors, which interfere with single-strand DNA break repair, exacerbating radiation-induced damage [59]. These inhibitors are particularly relevant for MGMT-unmethylated tumors, where TMZ resistance limits therapeutic options [12]. Although preclinical evidence shows synergy between PARP inhibitors and RT, their clinical efficacy is still under evaluation [49]. A recent randomized phase II trial (VELIPARIB with Radiation Therapy and Temozolomide in Unmethylated MGMT Glioblastoma; VERTU) demonstrated that the PARP inhibitor veliparib, combined with TMZ and RT, improved PFS in MGMT-unmethylated patients (PFS-6m 46% vs. 31% in standard arm), pending phase III confirmation [39]. RT-induced lymphopenia restricts synergy with immunotherapy, but MGMT methylation identifies patients with favorable immune microenvironments, and PT may reduce lymphopenia to enhance combined therapies [44]. Genomic profiling is a promising approach to personalize RT and its combination with targeted therapies or immunotherapy, with EANO 2025 guidelines recommending systematic profiling to optimize diagnosis and identify candidates for personalized trials [58,60].

## 5. Advances in the Treatment of Glioblastoma Multiforme

GBM is an aggressive brain tumor with a high recurrence rate. In non-elderly patients with good functional status (KPS ≥ 70), the Stupp protocol has been the standard of care for nearly two decades [12]. However, tumor resistance has limited progress over the past decade [36]. Current useful methods emphasize a multimodal strategy: maximal safe resection is followed by RT (60 Gy/30 fractions) with concurrent/adjuvant TMZ as the standard, improving OS especially in MGMT-methylated patients; TTFields are added to maintenance TMZ for survival benefit in selected cases; short-course RT (± TMZ) is leveraged for the elderly/frail; and at recurrence, options like re-resection, reirradiation, or systemic therapies (e.g., lomustine; bevacizumab for symptom control) are used. Investigational approaches like immunotherapies and targeted therapies show promise but lack phase III confirmation, emphasizing the need for randomized controlled trials (RCTs) to validate efficacy.

Integration of advanced medical imaging and nuclear medicine is essential for therapeutic decision-making and personalized GBM management, while supportive and palliative care are crucial throughout.

Below, we discuss promising methods to improve tumor control and quality of life while reducing toxicity, prioritizing RCTs where available.

### 5.1. Technological Advances in Therapies

Technological advancements have enhanced the precision of treatment, minimizing damage to healthy tissues [61]. PT is increasingly relevant, particularly in reirradiation and selected cases [61]. It leverages the Bragg peak, depositing maximum energy at the end of its range with a sharp fall-off, to target the tumor, with a relative biological effectiveness (RBE) of 1.1 [61]. A phase II RTC (NCT01854554) demonstrated that PT (60 Gy/30 fractions) significantly reduced grade ≥ 3 lymphopenia (14% vs. 39%), minimized irradiated brain volumes (V5–V40), and preserved immune function, potentially enhancing immunotherapy efficacy [6,44]. Although it did not delay cognitive decline compared to IMRT, it reduced fatigue and grade ≥ 2 toxicity, lowering doses to critical structures, suggesting potential for cognitive preservation in low-grade gliomas [6,62]. The NRG-BN001, a phase II RTC (NCT02179086) evaluating dose-escalated photon RT (75 Gy in 30 fractions) versus proton radiotherapy with TMZ, reports improved PFS but no mOS benefit (18.7 months) for the photon arm, with proton arm results pending [63,64]. Accessibility barriers, including high costs and limited centers, restrict PT adoption. Carbon ion radiotherapy (CIRT) achieves a mOS of 18 months in nGBM from retrospective studies [37]. Boron neutron capture therapy (BNCT) reports an mOS of 25.7 months in nGBM with TMZ and 18.9 months in rGBM from non-randomized studies [65,66], with accessibility limited by infrastructure and toxicity concerns (e.g., cerebral edema). Magnetic resonance-guided radiotherapy (MRgRT) achieves an mOS of 18.5 months and a PFS of 11.6 months in nGBM (UNITED, non-randomized phase II), with UNITED2 ongoing [67,68]. TTFields, a standard in NCCN (category 1) and American Society of Clinical Oncology (ASCO) guidelines but not endorsed by the National Institute for Health and Care Excellence (NICE) due to cost-effectiveness, low adherence, and biases in EF-14 (unblinded, selected patients), achieve an mOS of 20.9 months (PFS 6.7 months) in nGBM (phase III EF-14) and an mOS of 10.3 months in rGBM (EF-11). These biases include lack of blinding and selection of healthier patients, potentially inflating efficacy estimates. Trials include phase II 2-THE-TOP (NCT03405792), reporting an mOS of 24.8 months and PFS of 12.0 months with pembrolizumab plus TMZ in nGBM [14], and ongoing phase III trials (TRIDENT with RT/TMZ, EF-41 with TMZ plus pembrolizumab in nGBM) [69,70,71]. Modulated electrohyperthermia (mEHT), laser interstitial thermal therapy (LITT), and magnetic hyperthermia (MHT) have limited evidence in rGBM, supported by non-randomized phase I/II trials [72,73,74,75]. Table 6 details these modalities, emphasizing RCTs where available and noting accessibility barriers. Most of these modalities’ non-randomized or retrospective data limit comparisons with photon-based RT.

### 5.2. Modified Fractionation Schedules

Modified fractionation schedules optimize RT by counteracting tumor repopulation and reducing treatment duration [36,76]. Hypofractionation, standard in elderly patients with poor functional status (40 Gy/15 fractions), shows promise in younger patients (50–60 Gy/20 fractions with TMZ). The randomized phase II HART-GBM trial achieved an mOS of 26.5 months compared to 22.4 months with standard fractionation though no phase III trials confirm these findings [76]. An institutional study reported an mOS of 19.8 months [77], and a meta-analysis showed a 12-month OS of 71.3% across various ages [78]. Hyperfractionation shows no clear benefit, with similar survival outcomes and moderate toxicity compared to standard fractionation [36,79]. Dose escalation (75 Gy/30 fractions) improves PFS but not mOS (18.7 months) in the NRG-BN001 phase II RTC with TMZ in nGBM; PT arm results pending [63]. Dose escalation with a CIRT boost (16.8–24.8 GyE) achieves an mOS of 18 months [37]. Toxicities include radionecrosis (6.7–14.2%) and cognitive decline [77,80,81]. Tumor heterogeneity and hypoxia necessitate phase III trials, as detailed in Table 7.

### 5.3. Reirradiation

Reirradiation is a viable option for rGBM in patients with recurrence more than 6 months after initial RT, following multidisciplinary discussion. The 2025 ESTRO/EANO guidelines (KPS > 60; tumor volume < 35 cm^3^) and ASTRO 2025 guidelines (KPS ≥ 70; tumor volume ≤ 6 cm^3^) endorse its selective use [11,82]. Post-contrast T1 MRI delineates the GTV, complemented by [18F]-FET or [18F]-FDOPA PET to detect recurrence [82,83]. Regimens such as hypofractionated RT (35 Gy/10 fractions) with bevacizumab (BEV) achieve an mOS of 10.1 months and PFS of 3–6 months, with approximately 5% radionecrosis [82]. NRG Oncology/RTOG 1205 demonstrated improved PFS with hypofractionated RT plus BEV [82]. The LEGATO trial evaluates lomustine with or without reirradiation [84]. Hypofractionated stereotactic RT (25 Gy/5 fractions) showed outcomes similar to 35 Gy/5 fractions, with lower toxicity but higher radionecrosis in larger volumes [85]. CIRT (45 Gy RBE/15 fractions) improves mOS (8.0 months) compared to photons [37]. PT achieves an mOS of 7.8–19.4 months in reirradiation [64]. While not detailed here, brachytherapy, pulsed low-dose-rate radiotherapy (pLDR), and flash radiotherapy are promising for recurrent GBM [11,36]. See Table 8.

### 5.4. Neoadjuvant Therapy

Neoadjuvant therapy (NAT) reduces tumor volume to facilitate resection or enhance CRT, though it faces challenges such as the need for invasive stereotactic biopsy to confirm diagnosis [86]. In preoperative NAT, BEV in patients with low KPS improves resection (>95%), with an mOS of 15.7 months [87]. POBIG (phase I) evaluates stereotactic RT (6–14 Gy/1 fraction) [88], and PARADIGMA (phase II, NCT03480867) explores RT plus TMZ [86]. In postoperative NAT for unresectable GBM, TMZ plus BEV, the most promising regimen, achieves an mOS of 12.5 months and PFS of 7.4–8.6 months, but BEV increases intracranial hemorrhages [86,89,90]. For resectable GBM, the MAGMA phase III RCT (NCT02394639) evaluates neoadjuvant TMZ (75 mg/m^2^ daily) and extended adjuvant TMZ (150–200 mg/m^2^ until progression) before CRT (60 Gy/20 fractions), achieving an mOS of 23 months in MGMT-methylated cases, with hematological toxicity as a primary concern [91].

Neoadjuvant immunotherapies, such as pembrolizumab in resectable rGBM and triple immunotherapy (nivolumab plus ipilimumab plus relatlimab) in nGBM, show potential in selected subgroups [92,93]. Preliminary data from a single nGBM case in the GIANT trial (NCT06816927) suggest no recurrence at 17 months, pending further validation [93]. Future trials should combine immune checkpoint inhibitors (ICIs) with intratumoral oncolytic viruses to enhance immune responses, with strict patient selection due to variable toxicity [86]. ICI immunotherapy combined with oncolytic viruses may benefit selected subgroups [86] (Table 9)

### 5.5. Immunotherapy, Targeted Therapies, and Chemotherapy

Several immunotherapy trials, notably CheckMate 143, 498, and 548, have failed to demonstrate OS benefits, highlighting challenges in overcoming GBM’s immunosuppressive microenvironment [3,92,94]. Chemotherapies, targeted therapies, and immunotherapies are evaluated for newly diagnosed (nGBM) and recurrent GBM (rGBM) (Table 10).

Standard TMZ chemotherapy, established by the EORTC/NCIC CE.3 trial, is the reference treatment in nGBM, improving survival compared to RT alone [12], though intensive TMZ offers no benefit (RTOG 0525) [95]. In rGBM, metronomic TMZ shows limited activity, particularly after BEV (RESCUE), and lomustine offers modest results (EORTC 26101) [97,98,99]. Lomustine plus TMZ achieves an mOS of 48.1 months in nGBM with MGMT methylation (CeTeG/NOA-09) [96]. Chemotherapy- and RT-induced lymphopenia, as discussed in Section 4, may limit subsequent immunotherapy efficacy.

Targeted therapies, such as BEV, do not extend OS in nGBM (AVAglio, RTOG 0825; mOS 16.8 months; mPFS 10.6 months) but improve disease control in rGBM (BRAIN, EORTC 26101, BELOB; mOS 9.2 months; mPFS 4.2 months) [98,99,100,101] (see Table 9 for its neoadjuvant role). PARP inhibitors, such as veliparib, tested in the VERTU trial, and niraparib (under evaluation in phase 3 trials for MGMT-unmethylated nGBM) are a promising targeted therapy in nGBM with unmethylated MGMT by enhancing radiation-induced damage, one of the few strategies with clinical evidence in specific subgroups [39]. Antibody-drug conjugates (ADCs) targeting EGFR are also emerging in early-phase trials for rGBM. Cilengitide is ineffective in nGBM (CENTRIC, CORE) [99,102], as are erlotinib and everolimus, both inhibitors of tumor signaling pathways [41,99]. Regorafenib has limited efficacy in rGBM (REGOMA), outperforming erlotinib or everolimus but with modest benefits [99]. BRAF/MEK inhibitors show promising responses in BRAF V600E-mutated cases [99]. IDH inhibitors, such as vorasidenib, are under investigation for IDH-mutant gliomas, with no data in GBM [99].

In immunotherapy, nivolumab does not outperform BEV in rGBM (CheckMate 143) or TMZ with RT in nGBM with unmethylated (CheckMate 498) or MGMT-methylated (CheckMate 548), where treatment-induced lymphopenia may reduce efficacy [3,92,94]. Adoptive cellular therapies, such as CAR-T, and oncolytic viruses, such as G47Δ (modified herpes simplex virus) or DNX-2401 (adenovirus), are promising in rGBM. DNX-2401, in monotherapy (20% 3-year survival) or combined with pembrolizumab in the CAPTIVE trial (mOS 12.5 months, ongoing), shows preliminary efficacy, though immunosuppression from dexamethasone and RT may limit its effectiveness [92,105]. Neoadjuvant pembrolizumab before surgery in rGBM improves survival with an mOS of 13.8 months in specific subgroups [92,104] (see Table 9 for neoadjuvant use). The DCVax-L vaccine benefits nGBM (mOS 19.3 months) and rGBM (mOS 13.2 months) but is not standard; rindopepimut does not improve survival in EGFRvIII-positive nGBM (ACT IV) [92,106]. Other vaccines are in development for GBM [92]. Cytokines like interferon-alpha (IFN-α) enhance TMZ in nGBM (mOS 26.7 months, phase III), though not standard, and neoadjuvant triple immunotherapy before surgery prevents recurrences in isolated nGBM cases (GIANT, ongoing) [3,93]. The blood–brain barrier (BBB) and treatment-induced immunosuppression limit the combination of immunotherapy with RT [92] (Table 10)

### 5.6. Advanced Imaging and Theranostics

Multiparametric MRI, including T1/T2-FLAIR, dynamic susceptibility contrast perfusion (DSC), diffusion-weighted imaging (DWI), and magnetic resonance spectroscopy (MRS), detects gliomas with high sensitivity, evaluates recurrence per RANO 2.0 criteria, including non-enhancing disease in IDH-mutant gliomas, and distinguishes pseudoprogression from true progression using DSC, DWI, and MRS (Table 11) [47,107,108], further enhancing early GBM diagnosis and personalized RT planning by detecting non-enhancing tumors with MRI/PET and optimizing tumor control probability using radiobiological models like Poisson to tailor dose prescriptions [48,109].

Challenges include high costs and the need for specialized expertise in interpreting multiparametric MRI [107,108]. PET with amino acid tracers ([18F]-FET, [11C]-MET, [18F]-FDOPA, [18F]-FACBC) and [68Ga]-PSMA-11 differentiates recurrence from pseudoprogression with high specificity [110,111,112,113]. These techniques optimize RT planning by improving tumor delineation and enabling dose escalation in high-risk regions. A phase I trial using multiparametric MRI and [18F]-FDOPA PET achieved an mOS of 23 months without significant toxicity, though phase III trials have not confirmed OS benefits [48]. Theranostics with [131I]-IPA achieves an mOS of 16 months in rGBM, limited by BBB penetration [111] (Table 11). Ongoing trials are evaluating novel theranostic agents [111,113]. Alpha particle-emitting radiopharmaceuticals, delivering high-LET radiation via ligands such as substance P targeting hypoxic tumor cells, show promise in preclinical and early clinical studies for GBM with minimal damage to surrounding healthy tissue due to their short range and high cytotoxicity but remain investigational [38].

### 5.7. Artificial Intelligence in GBM Radiotherapy

Artificial Intelligence (AI) revolutionizes GBM RT by integrating multiparametric MRI and PET data to enhance treatment planning, predict recurrence, and personalize therapy through advanced radiomic analysis [48,109,114]. AI-driven algorithms automate tumor segmentation, quantify biological heterogeneity, and differentiate true progression from pseudoprogression, improving target volume delineation and adaptive dose strategies [48,114]. By leveraging radiobiological models like Poisson, AI optimizes Tumor Control Probability (TCP) calculations, tailoring doses to tumor characteristics and biomarkers such as MGMT methylation [109,114]. Challenges include data variability, non-standardized imaging protocols, and limited model interpretability, necessitating multicenter prospective trials to validate clinical efficacy and reproducibility of AI-driven TCP optimization [48,114].

## 6. Conclusions

Despite significant advancements in GBM management, RT continues to face inherent limitations, primarily due to diffuse tumor infiltration and intrinsic radioresistance, often driven by specific molecular alterations like EGFR amplification, MGMT methylation, and TERT promoter mutations, alongside challenges from the immunosuppressive microenvironment and cancer neuroscience.

Advanced RT techniques, including IMRT, PT, CIRT, BNCT, TTFields, and localized therapies like MHT and LITT for rGBM, optimize dose precision, yet recurrences remain common.

Promising strategies such as neoadjuvant approaches, modified fractionation like hypofractionated RT, and reirradiation for recurrent GBM show therapeutic potential.

The integration of systemic therapies, including immunotherapies, chemotherapies, and targeted agents, is crucial for enhancing tumor control, though RT-induced lymphopenia limits immunotherapy efficacy. Multiparametric MRI and PET enable tailored RT planning, while radiotheranostics like [131I]-IPA broaden treatment options, despite BBB challenges. To advance GBM management, phase III RCTs are needed to validate dose-escalation strategies with PT, biomarker-driven trials to personalize therapies, and AI integration for adaptive RT planning to differentiate true progression from pseudoprogression.

## 7. Future Directions

The future of GBM radiotherapy lies in leveraging molecular biomarkers and advanced imaging (e.g., multiparametric MRI; amino acid PET) to enable tailored treatment planning. Biomarker-driven clinical trials are critical to personalize RT and systemic therapies, integrating molecular profiles to overcome radioresistance. Phase III RCTs are urgently needed to validate PT dose escalation, with the phase II NRG-BN001 trial (NCT02179086) designed to assess a threshold for advancing to phase III, though preliminary photon arm results did not meet criteria, and proton arm results are pending. Other phase III RCTs, such as TRIDENT (NCT04471844) and EF-41 (NCT06556363) for TTFields with systemic therapies, and LEGATO (NCT04078568) for reirradiation, aim to confirm clinical benefits. Emerging trials, including the phase 3 Gliofocus (NCT06388733) comparing niraparib vs. TMZ in newly diagnosed MGMT-unmethylated GBM, may address TMZ challenges in resistant subgroups [103]. According to the 2025 SNO-EANO consensus, strategies to overcome the BBB, such as focused ultrasound or local delivery, and liquid biopsy technologies to monitor tumor evolution are promising to enhance therapeutic delivery and personalization [45].

AI-driven predictive models can achieve precise, adaptive RT planning by integrating tumor biology and imaging data, enhancing differentiation of true progression from pseudoprogression and optimizing dose delivery. Ethical considerations, such as mitigating biases in AI training data, are crucial for clinical adoption.

Radiotheranostics, including [131I]-IPA and agents like [177Lu]-PSMA, and alpha particle-emitting radiopharmaceuticals for targeted therapy with high LET to address hypoxia, hold promise for targeted imaging and therapy, improving oncological outcomes and quality of life by overcoming blood–brain barrier challenges, with phase I/II trials showing improved PFS and potential synergy with RT to elevate median mOS in rGBM.

## Figures and Tables

**Table 1 biomedicines-13-02136-t001:** Survival patterns in glioblastoma multiforme by MGMT methylation status and therapy.

Intervention	Median Survival–Unmethylated	Median Survival–Methylated	Conclusion
Surgery + RT	11.8 months	15.3 months	RT alone improves OS in methylated tumors, but the benefit is limited.
Surgery + RT + TMZ	12.7 months	21.7 months	The addition of TMZ significantly increases OS in methylated tumors (*p* = 0.007).
Surgery + RT + TMZ + TTFields	16.9 months	31.6 months	The combination of TTFields with RT and TMZ offers the greatest OS benefit, especially in methylated tumors.

Legend: RT: Radiotherapy; TMZ: Temozolomide; TTFields: Tumor Treating Fields; OS: Overall Survival. Adapted from Roubil JG et al. [17], with data from Stupp et al. [12,16].

**Table 2 biomedicines-13-02136-t002:** Summary of studies on hypofractionation in patients with GBM.

Study	Patients	Treatment	Results	Conclusion
Roa et al. (2004) [19]	N = 100 ≥60 y.o. KPS ≥ 50.	RT (60 Gy/6 weeks) vs. RT (40 Gy/3 weeks).	- OS comparable (5.1 vs. 5.6 months) - Better tolerance - Lower use of post-treatment steroids in short RT	Short RT is effective and more comfortable
Perry et al. (2017) [18]	N = 562 ≥65 y.o.	RT (40 Gy/15 fr) ± concomitant and adjuvant TMZ.	- Better OS with combination (9.3 mo vs. 7.6 m), especially in MGMT-methylated - More hematological adverse effects with TMZ	Short RT + TMZ improves OS. Standard in eligible patients
Malmström et al. Nordic Trial (2012) [22]	N = 342 ≥60 y.o. WHO 0–2	TMZ vs. RT 60 Gy/6 weeks vs. RT 34 Gy/2 weeks.	- OS: TMZ 8.3 m/RT 6.0 mo/hypofractionated RT 7.5 mo - greater benefit in >70 years - MGMT-methylated: predictive marker of response to TMZ (OS 9.7 mo vs. 8.2 in RT)	- TMZ and hypofractionated RT > standard RT - TMZ vs. hypofractionated RT in patients >70 years old, depending on MGMT-methylated. Individualized management
Roa et al. IAEA Trial (2015) [20]	N = 98 ≥65 y.o. and/or fragile	RT (25 Gy/5 fr) vs. RT (40.05 Gy/15 fr).	Similar OS (7.9 vs. 6.4 mo).	Hypofractionated RT is feasible and effective in frail patients
Minniti et al. (2009) [21]	N = 43 ≥70 y.o. KPS ≥ 60	RT 30Gy/6 fr vs. adj. TMZ	- OS 9.3 mo - PFS 6.3 mo - Best in KPS > 70 - Acceptable toxicity	Combination is effective and safe in selected patients with limited prognosis

Legend: RT: Radiotherapy; TMZ: Temozolomide; KPS: Karnofsky Performance Status; WHO: World Health Organization scale; OS: Overall Survival; PFS: Progression-Free Survival; mo: Months; fr: Fractions.

**Table 3 biomedicines-13-02136-t003:** Summary of radiotherapy volumes for GBM.

Guideline/Study	GTV	CTV	PTV	Edema Inclusion
EORTC (Stupp) [12]	Tumor + cavity	CTV1: GTV + edema + 20 mm CTV2: GTV + 25 mm	PTV = CTV + 3–5 mm	Yes, included in initial phase
RTOG/NRG (2019) [14]	Phase 1: tumor + edema; Phase 2: tumor + cavity	CTV1: GTV1 + 20 mm CTV2: GTV2 + 20 mm	PTV = CTV + 3–5 mm	Yes, included in initial phase
ESTRO-EANO 2023 [25]	Surgical cavity + post-surgical T1 enhancement	GTV + 15 mm (adjusted to anatomy)	CTV + individual margin (usually ≤3 mm with IGRT)	Not systematically included
ESTRO-ACROP 2016 [26]	Cavity + residual tumor	GTV + 15–20 mm, adjusted to anatomical barriers	CTV + 3–5 mm	Not systematically included
Minniti et al. (2010) [28]	Surgical cavity + post-surgical T1 enhancement	CTV1: GTV + 2 cm CTV2: GTV + 1 cm	CTV + 3mm	Not included
Chang et al. (2007) [29]	Cavity + T1 tumor	CTV1: GTV + 20 mm CTV2: GTV + 5 mm	CTV + 5 mm	Not included
ASTRO 2025 (1-phase) [11]	Surgical cavity + post-surgical T1 enhancement	GTV + 10–20 mm (adjusted to anatomy, including edema is optional)	PTV = CTV + 3–5 mm	Optional.
ASTRO 2025 (2-phase) [11]	Phase 1: cavity + T1 + T2/FLAIR enhancement Phase 2: cavity + T1 enhancement	CTV1: GTV1 + 10–20 mm CTV2: GTV2 + 10–20 mm (adjusted to anatomy)	PTV = CTV + 3–5 mm	Yes, included in initial phase; not in phase 2
MDACC Kumar et al. (2020) [27]	Cavity + T1 enhancement	Initial GTV + 2 cm, boost GTV + 5 mm	CTV + 5 mm	Not included

Legend: GTV: Gross Tumor Volume; CTV: Clinical Target Volume; PTV: Planning Target Volume; IGRT: Image-Guided Radiotherapy; MDACC: MD Anderson Cancer Center.

**Table 4 biomedicines-13-02136-t004:** Limiting factors of radiotherapy in the treatment of GBM.

Category	Limitation	Description/Evidence
Technical	Local recurrence	80–90% of recurrences occur within 2 cm of the irradiated field due to diffuse infiltration, even with IMRT, VMAT, or proton therapy [32].
	Dosimetric constraints	Limits such as <54 Gy to the brainstem and optic chiasm restrict dose escalation; proton therapy minimizes irradiated volumes [25].
	Acute and late toxicity	Fatigue, radiation necrosis, and cognitive deficits; brain volumes (V20Gy, V40Gy) increase neurotoxicity, reducible with IMRT, VMAT, and proton therapy [25].
	Delay in RT initiation	Delays >6 weeks worsen OS and PFS; moderate delays (~6 weeks) may benefit patients with residual disease [33,34].
	Cost and adherence of new technologies	Tumor Treating Fields (TTFields) extend PFS by 2.7 months (6.7 vs. 4.0 months), limited by cost and adherence (≥18 h/day) [35].
Biological	Tumor infiltration	The diffuse nature of GBM allows tumor cells to escape the radiation field [32].
	Tumor hypoxia	Tumor hypoxia, by activating HIF-1α, reduces RT efficacy by promoting cell survival [36]. Mitigated with modulated electrohyperthermia (mEHT) or hypoxia-sensitizing agents; high-linear energy transfer (LET, energy deposited per unit length) radiation (e.g., carbon ions; alpha particles) is promising but investigational [37,38].
	Cellular radioresistance	Tumor stem cells and pathways like MGMT (O6-methylguanine-DNA methyltransferase), EGFR amplification (57% per TCGA [The Cancer Genome Atlas]), PTEN mutations (40% per TCGA), and CDKN2A/B deletions (60% per TCGA) drive resistance; PARP inhibitors (e.g., veliparib) and other radiosensitizers show promise by inhibiting DNA repair [36,39,40]. See Table 5 for additional molecular determinants.
	EGFR amplification	In 57% per TCGA, activates PI3K/Akt and RAS/RAF/MAPK, conferring resistance; PTEN mutations (40% per TCGA) enhance this pathway; inhibitors like erlotinib have limited benefits [36,40,41].
	SVZ as a reservoir	The SVZ (subventricular zone), with mutated stem cells (TERT promoter mutation >70% per TCGA, PTEN, TP53, EGFR), drives regrowth; irradiating the SVZ with doses ≥56 Gy (ipsilateral) or ≥50 Gy (contralateral) does not improve PFS or OS [40,42,43].
	RT-induced lymphopenia	Extensive irradiation causes grade 3+ lymphopenia (14% with protons vs. 39% with photons), limiting immunotherapy efficacy [44].
	Microenvironment and cancer neuroscience	Immunosuppressive microenvironment (e.g., macrophages/microglia supporting tumor growth) and neuronal crosstalk (e.g., synapses promoting proliferation/invasion) enhance resistance [45].
Clinical	Pseudoprogression	Affects 30-40% of patients with methylated MGMT after TMZ, complicating radiological assessment up to 12 weeks [46,47].
	Lack of clinical impact of biomarkers and advanced imaging	Multimodal imaging (multiparametric MRI; amino acid PET) enables RT personalization, achieving an mOS of 23 months in a phase I trial [48]. Biomarkers (EGFR, PTEN, and TERT) allow patient stratification but, except for MGMT-methylated, do not improve OS in phase III trials [49].
	Lack of consistent benefits from combined therapies	Targeted therapies and immunotherapies do not improve OS, though they extend PFS in nGBM [50].

Legend: PFS: Progression-Free Survival; mOS: Median Overall Survival; IMRT: Intensity-Modulated Radiotherapy; VMAT: Volumetric Modulated Arc Therapy; PT: Proton Therapy; MGMT: O6-Methylguanine-DNA Methyltransferase; EGFR: Epidermal Growth Factor Receptor; PTEN: Phosphatase and Tensin Homolog; TERT: Telomerase Reverse Transcriptase; TP53: Tumor Protein p53; SVZ: Subventricular Zone; HIF-1α: Hypoxia-Inducible Factor 1-Alpha; nGBM: Newly Diagnosed Glioblastoma Multiforme; RT: Radiotherapy; TMZ: Temozolomide; MRI: Magnetic Resonance Imaging; PET: Positron Emission Tomography. TCGA: The Cancer Genome Atlas.

**Table 5 biomedicines-13-02136-t005:** Molecular biomarkers in RT for GBM.

Biomarker	Frequency (IDH-wt)	Impact on RT	Therapeutic Status
MGMT-methylated	~45%	Greater sensitivity to RT + TMZ (beneficial prognostic marker)	Standard treatment with TMZ. ESCAT I [12,51,52]
IDH1/2 Mutation	0% (defines IDH-wt)	Greater radiosensitivity and better prognosis in IDH-mutant astrocytoma grade 4 (rare, ~10% of grade 4 gliomas)	Favorable stratification. ESCAT I [53]
Amplified EGFR	57% (per TCGA)	Activates PI3K/AKT; resistance to RT	Inhibitors without relevant clinical efficacy. ESCAT IIIA [54]
PTEN mutation/loss	~40% (per TCGA)	PI3K/AKT pathway; promotes resistance to RT	No approved effective therapies [40,55]
PIK3CA mutation	~10% (per TCGA)	Stimulates cellular survival signals	No approved effective therapies [40,55]
Amplified CDK4/6	~15% (per TCGA)	Stimulates cell cycle progression	Inhibitors under clinical study [40,55]
CDKN2A/B deletion	~50% (per TCGA)	Loss of cell cycle control; poor prognosis	No effective targeted therapies [49,56]
TERT mutation	>70% (per TCGA)	Uncertain impact; possible role in immune evasion	Under investigation as an immunotherapeutic target [3,57]

Legend: IDH: Isocitrate Dehydrogenase; wt: Wild-Type; MGMT: O6-Methylguanine-DNA Methyltransferase; RT: Radiotherapy; TMZ: Temozolomide; ESCAT: ESMO Scale for Clinical Actionability of Molecular Targets; EGFR: Epidermal Growth Factor Receptor; PTEN: Phosphatase and Tensin Homolog; PI3K/AKT: Phosphatidylinositol 3-Kinase/Protein Kinase B; CDK4/6: Cyclin-Dependent Kinases 4 and 6; TERT: Telomerase Reverse Transcriptase; CDKN2A/B: Cyclin-Dependent Kinase Inhibitor 2A/B; TCGA: The Cancer Genome Atlas.

**Table 6 biomedicines-13-02136-t006:** Technological advances in therapies against GBM.

Technique	Description	mOS/PFS	Evidence	Results Pending	Limitations
PT	Focused dose delivery (Bragg peak, spread-out Bragg peak [SOBP], intensity-modulated proton therapy [IMPT], RBE 1.1)	nGBM: mOS 21–24 months; PFS 6.6–8.9 months	Phase II (NCT01854554, *n* = 67; PT vs. XRT); reduces grade ≥ 3 lymphopenia (14% vs. 39%, *p* = 0.024), fatigue (24% vs. 58%, *p* = 0.05), toxicity grade ≥ 2 (0.35 vs. 1.15, *p* = 0.02), V5–V40 [44,62,63,64]	NRG-BN001 (NCT02179086): Phase II trial, photon arm (75 Gy/30 fractions) improves PFS, mOS 18.7 months; proton arm results pending, in nGBM [63,64]	Cost, accessibility
CIRT	High energy transfer, RBE 2.5–5	nGBM: mOS 18 months; rGBM: mOS 8 months	nGBM: CIRT boost (18 GyE/6 fx) or with TMZ (retrospective randomized phase II); rGBM: 45 GyE/15 fx (non-randomized comparative) [37]	CINDERELLA (NCT01166308): Phase I/II, CIRT vs. FSRT in rGBM; CLEOPATRA (NCT01165671): Phase II randomized, CIRT vs. proton boost in nGBM [37]	Cost, limited centers. *
BNCT	Selective damage with boron-10 (L-BPA, BSH); planning with 18F-BPA PET	nGBM: mOS 25.7 months (with surgery + TMZ); rGBM: mOS 18.9 months	nGBM: Surgery, BNCT (~40 Gy-Eq) and TMZ, without conventional RT; rGBM: Non-randomized Phase II (JG002), minimum 39.8 Gy-Eq [65,66]	Under investigation, non-randomized [65,66]	Toxicity (cerebral edema, hyperamylasemia, alopecia); infrastructure. *
MRgRT	Daily adaptation with T1/T2 MRI	nGBM: mOS 18.5 months, PFS 11.6 months (long course); marginal failure 4.1%	Non-randomized Phase II UNITED (NCT04726397, *n* = 98; CTV 5 mm, 60 Gy/30 fx) [67,68]	UNITED2 (NCT05565521): Phase II non-randomized, 40 Gy/15 fx + boost 52.5 Gy/15 fx, PFS at 6 months [67]	Cost, evidence in development. *
TTFields	Alternating electric fields (200 kHz, 1–3 V/cm); with TMZ + RT	nGBM: mOS 20.9 months, PFS 6.7 months (EF-14) rGBM: mOS 10.3 months (EF-11) nGBM: mOS 24.8 months, PFS 12.0 months, 1-year survival 82.61% (2-THE-TOP)	Phase III EF-14 (NCT00916409, *n* = 695; TTFields+TMZ in nGBM, HR 0.63, *p* < 0.001); phase III EF-11 (NCT00379470, TTFields in rGBM); phase II 2-THE-TOP (NCT03405792, TTFields + TMZ + pembrolizumab in nGBM) [69,70,71]	Phase III TRIDENT (NCT04471844): RT/TMZ in nGBM; EF-41 (NCT06556363): TMZ+pembrolizumab + TTFields in nGBM [69,70,71]	Cost, adherence, dermatitis; NICE does not endorse it due to cost-effectiveness, EF-14 biases (unblinded, selected patients)
mEHT	Thermal radiosensitization (13.56 MHz, 40–43 °C); immunogenic potential	nGBM: 1-year survival 73.33%; rGBM: mOS 7.7 months, 1-year survival 37.33%	Observational studies 2006–2018 (*n* = 450); 1 nGBM study, 6 rGBM studies (with ddTMZ); phase I for safety; no RCTs [72,73,74]	In research, non-randomized [72,73,74]	Weak evidence, not in guidelines, few centers. *
LITT	MRI-guided laser thermal ablation	rGBM: mOS ~8–12 months	Phase I/II in rGBM; comparable to re-surgery in unifocal lobar rGBM; no RCTs [75]	In research, non-randomized [75]	Small focal lesions, no RCTs, post-procedural edema. *
MHT	AMF-guided hyperthermia with magnetic nanoparticles (40–45°C)	rGBM: mOS not reported	Phase I/II in rGBM with RT; Proven safety and feasibility; no RCTs [75]	In research, non-randomized [75]	No RCTs, technical challenges (MNP, thermometry), few centers. *

Legend: AMF: Alternating magnetic field; BSH: Sodium borocaptate; CIRT: Carbon ion radiotherapy; ddTMZ: Dense dose temozolomide; FSRT: Fractionated stereotactic radiotherapy; GyE: Gray equivalent; HR: Hazard ratio; IMPT: Intensity-modulated proton therapy; L-BPA: L-4-boronophenylalanine; LITT: Laser interstitial thermal therapy; mEHT: Modulated electrohyperthermia; MHT: Magnetic hyperthermia; MNP: Magnetic nanoparticles; MRgRT: Magnetic resonance-guided radiotherapy; RBE: Relative biological effectiveness; RCTs: Randomized clinical trials; SOBP: Spread-out Bragg peak; V5–V40: Volume of tissue receiving 5 to 40 Gy of radiation; XRT: Photon radiotherapy. * Note: non-randomized data limit comparisons with photon-based radiotherapy.

**Table 7 biomedicines-13-02136-t007:** Modified fractionation schedules.

Schedule/ Description/ Indications	mOS/PFS	Evidence	Ongoing Trials	Limitations
Hypofractionation: 50–60 Gy/20 fractions with TMZ, primarily in younger patients (≤65 years, KPS ≥ 70)	mOS: 26.5 months, PFS: 13.2 months [76]; mOS: 19.8 months, PFS: 7.7 months [77]; 12-month OS: 71.3%, 12-month PFS: 40.8% (various ages) [78]	HART-GBM trial (phase II, 60 Gy/20 fx vs. 60 Gy/30 fx, *n* = 83, with TMZ, patients aged 16–65 years) [76]; Institutional study (50 Gy/20 fx vs. 60 Gy/30 fx, *n* = 41, with TMZ, patients < 65 years) [77]; Meta-analysis (*n* = 484, phase I/II and retrospective, various ages) [78]; Meta-analysis (n not specified, phase II/III, HFRT vs. CFRT, various ages) [80]	SAGA (NCT05781321, randomized phase II, 5–10 fx photons guided by [18F]-FDOPA PET, evaluating survival, cost-effectiveness, and failure patterns, patients ≥ 18 years, ClinicalTrials.gov)	Grade ≥ 3 radionecrosis (6.7% HFRT, 7.7% CFRT), tumor heterogeneity [77,80]
Hyperfractionation: 37 × 1.6 Gy or 30 × 1.8 Gy bid with TMZ, experimental	No clear benefit (mOS 14.9 vs. 16.9 months, *p* = 0.26) [79]	Retrospective analysis (HFRT vs. NFRT, *n* = 484, with TMZ) [79]; Review (variable dose, with TMZ) [36]		Lack of efficacy; moderate toxicity [36,79]
Dose escalation: 75 Gy/30 fx (IMRT/PT) with TMZ or 16.8–24.8 GyE boost (CIRT), selected patients (KPS ≥ 70)	Photon arm: mOS: 18.7 months, improved PFS (60); mOS: 18 months (CIRT boost) [37]	NRG-BN001 phase II trial (75 Gy/30 fx vs. 60 Gy/30 fx, *n* = 299, with TMZ, photons, preliminary results improve PFS, mOS 18.7 months) [63]; CIRT retrospective study (16.8–24.8 GyE/8 fx after 50 Gy photons, *n* = 32) [37]	NRG-BN001 (PT arm, randomized phase II) [63]; CLEOPATRA (CIRT boost, randomized phase II) [37]	Grade ≥ 3 radionecrosis (up to 29%), tumor heterogeneity, no mOS improvement [63,81]

Legend: bid: Twice daily; CIRT: Carbon ion radiotherapy; GyE: Gray equivalent.

**Table 8 biomedicines-13-02136-t008:** Reirradiation.

Modality	Description	Indications	MOS/ PFS	Evidence	Trials	Limitations
HFRT/ CFRT	35 Gy/10 fx or 36 Gy/18 fx ± BEV (CTV: GTV + ≤5 mm, adjusted to anatomical barriers; PTV: CTV + ≤3 mm) [82]	KPS > 60, recurrence > 6 months, volume < 35 cm^3^ (ESTRO/EANO 2025) [82]; KPS ≥ 70, volume ≤ 6 cm^3^ (ASTRO 2025), multidisciplinary discussion [11]	mOS 7–12 months, PFS 3–6 months [82]	NRG Oncology/RTOG 1205 (phase II, HFRT + BEV, mOS 10.1 months) [82]	LEGATO (phase III, lomustine ± HFRT) [84]	Radionecrosis (~5%), lack of phase III trials [82]
SRS/ HSRT	SRS: 12–15 Gy/1 fx (CTV: GTV, usually without margin; PTV: CTV + 0–1 mm); HSRT: 25 Gy/5 fx (CTV: GTV, without margin; PTV: CTV + 3 mm) [82,83,85]	SRS: volume < 10 cm^3^; HSRT: volume ≤ 150 cm^3^ (median 55 cm^3^) [82,83,85]	mOS 9–11 months, PFS 5–6 months [82,83,85]	Retrospective trials (SRS, volume < 12.5 cm^3^) [82]; Phase II, HSRT 25 Gy/5 fx, mOS 9.2 months, PFS 4.9 months; 35 Gy/5 fx shows no improvement in PFS (4.9 vs. 5.2 months) or OS (9.2 vs. 10 months) [85]		Radionecrosis (<3.5% if volume <12.5 cm^3^ [82]; ~25% in HSRT [85]), phase II escalation from 25 Gy/5 fx to 35 Gy/5 fx does not improve PFS (4.9 vs. 5.2 months) or OS (9.2 vs. 10 months) [85]
CIRT/PT	CIRT: 45 Gy RBE/15 fx; PT: 33–46.2 Gy variable (CTV: GTV + ≤3 mm, adjusted to anatomical barriers; PTV: CTV + ≤3 mm) [37,64,83]	Selected patients [37,64,83]	mOS 7.8–19.4 months, PFS 5.5–13.9 months [37,64,83]	Retrospective studies (CIRT, PT) [83]; CIRT 45 Gy RBE/15 fx, mOS 8.0 months vs. photons [37]; PT 33–46.2 Gy, mOS 7.8–19.4 months, low toxicity [64]	CINDERELLA (phase I/II, CIRT vs. FSRT) [64]	Cost, accessibility, limited prospective data, toxicity not reported

Legend: BEV: Bevacizumab; CIRT: Carbon ion radiotherapy; fx: Fraction; HSRT: Hypofractionated stereotactic radiotherapy; PT: Proton therapy; RBE: Relative biological effectiveness; SRS: Stereotactic radiosurgery.

**Table 9 biomedicines-13-02136-t009:** Neoadjuvant therapy studies in GBM.

Preoperative Neoadjuvant Therapy for GBM				
Modality	Description	mOS/PFS	Trial/ Evidence	Limitations
Preoperative RT	SRS (6–14 Gy/1 fx)	Not reported	POBIG (phase I) [88]	No phase III trials
RT + Preoperative TMZ	RT + Preoperative TMZ	Not reported	PARADIGMA (phase II, NCT03480867) [86]	Pending results
BEV	BEV (10 mg/kg) preoperative	mOS: 15.7 months, PFS: 10.1 months	Miyake et al. (phase II, *n* = 12) [87]	Small sample size; limited data on toxicity
Pembrolizumab	Pembrolizumab (ICI, 200 mg every 3 weeks) pre-surgery	mOS: 13.7 months	(phase II, rGBM) [92]	Small sample size, immunological toxicity, dose heterogeneity, potential influence of steroids and bevacizumab
Triple Immunotherapy	Nivolumab + ipilimumab + relatlimab pre-surgery	No recurrence at 17 months (*n* = 1)	GIANT (phase I, nGBM, NCT06816927) [93]	Single case; preliminary data
**Postoperative Neoadjuvant Systemic Therapy Prior to Standard Chemoradiotherapy for Unresectable or Inoperable GBM**				
**Modality**	**Description**	**mOS/PFS**	**Trial** **/Evidence**	**Limitations**
TMZ + BEV	TMZ (75 mg/m^2^) + BEV (10 mg/kg) pre-RT	mOS: 12.5 months, PFS: 7.4–8.6 months	Bihan et al. (retrospective, *n* = 8) [89] Balana et al. (phase II, *n*= 102) [90]	Intracranial hemorrhages *; increased toxicity
**Postoperative Neoadjuvant Systemic Therapy Prior to Chemoradiotherapy for Resectable GBM**				
**Modality**	**Description**	**mOS/PFS**	**Trial/Evidence**	**Limitations**
TMZ	TMZ (75 mg/m^2^ daily) <7 days post-surgery and extended adjuvant (150–200 mg/m^2^ until progression), before CRT (60 Gy/20 fx in MAGMA)	mOS: 23 months, PFS: 11.5 months	Jiang et al. (retrospective, *n* = 375); MAGMA (phase III) [91] **	Hematological toxicity, MAGMA pending

Legend: BEV: Bevacizumab; fx: Fraction; SRS: Stereotactic radiosurgery. * Note: Intracranial hemorrhages primarily associated with BEV (4.2% in Balana et al.) [90]. ** Note: Greater benefit in MGMT-methylated cases (HR 0.60, Jiang et al.) [91].

**Table 10 biomedicines-13-02136-t010:** Immunotherapy, targeted therapies, and chemotherapy.

Category	Modality	mOS/PFS	Evidence	Notes
Chemotherapy (nGBM)	Standard TMZ	mOS 14.6 months, mPFS 6.9 months	EORTC/NCIC CE.3 [12]	Standard treatment
	Intensive TMZ	No improvement in mOS or PFS	RTOG 0525 [95]	Not recommended
	Lomustine + TMZ	mOS 48.1 months	CeTeG/NOA-09 [96]	MGMT-methylated
Chemotherapy (rGBM)	Metronomic TMZ	PFS-6 24% (1st recurrence); PFS-6 4.4% (post-BEV)	Phase II [97]	Limited efficacy, especially after bevacizumab
	Lomustine	mPFS 1.5 months	EORTC 26101 [98,99]	Limited efficacy; alone or with bevacizumab
Targeted Therapies (nGBM)	BVZ	mOS 16.8 months, mPFS 10.6 months	AVAglio, RTOG 0825 [99,100,101]	Not recommended as initial treatment; see Table 9 for neoadjuvant use
	Cilengitide + TMZ	No improvement SG/PFS	CENTRIC, CORE [99,102]	Not recommended
	PARP inhibitors (e.g., veliparib)	6-month PFS 46% (95% CI: 36–57%) vs. 31% (95% CI: 18–46%) in nGBM with unmethylated MGMT; no mOS benefit (12.7 vs. 12.8 months)	Phase II VERTU, preclinical synergy with RT/TMZ 36	Promising for 6-month PFS in nGBM with unmethylated MGMT, requires phase III confirmation, limited by tumor heterogeneity
	PARP inhibitor (niraparib) vs. TMZ	Emerging (phase 3 for MGMT unmethylated)	Phase 3 Gliofocus [103] (NCT06388733)	Ongoing; potential alternative to TMZ in unmethylated cases; PFS/OS endpoints
Targeted Therapies (rGBM)	BVZ	mOS 9.2 months, mPFS 4.2 months	BRAIN, EORTC 26101, BELOB [98,99,101]	Recommended in symptomatic relapse; see Table 9 for neoadjuvant use
	Regorafenib	mOS 7.4 months, mPFS 2.0 months	REGOMA [99]	Limited efficacy in rGBM
	BRAF/MEK inhibitors	Partial answers	Basket trials [99]	Compassionate use or use in clinical trials
	IDH inhibitors (vorasidenib)	No data in GBM	Phase I [99]	In research for IDH-mutant gliomas
	Erlotinib, Everolimus	Ineffective	Phase II [41,99]	Not recommended
	Antibody-drug conjugates (ADCs)	Emerging (e.g., phase I/II for EGFR-targeted)	Preclinical/early trials [45]	Promising for targeted delivery, but no phase III data; potential synergy with RT
IT (rGBM)	ICI: Nivolumab	mOS 9.8 months	CheckMate 143 [3,92]	No OS gain over BEV
	ICI: Pembrolizumab	mOS 13.8 months, mPFS 3.3 months, PFS-6 19.5%	Phase II (NCT02852655) [92,104]	Phase II, neoadjuvant to surgery; benefit in subgroups; see Table 9
	ACT: CAR-T, TILs, LAK	mOS 20.5 months, 1 RC	Phase I [92]	Phase I, preliminary data
	Vaccines: DCVax-L	mOS 13.2 months	Phase III [92]	Phase III, without RT; benefit in mOS, non-standard
	OV: DNX-2401, G47Δ, PVSRIPO	mOS 12.5-20.2 months	Phase I/II [92,105]	Preliminary results from CAPTIVE (DNX-2401 + pembrolizumab, ongoing)
	Cytokines: L19TNF + lomustine	mPFS 43.3 weeks	[92]	Phase I, preliminary data
IT (nGBM)	ICI: Nivolumab + RT	mOS 13.4 months	CheckMate 498 [94]	No OS gain over TMZ + RT; MGMT not methylated
	ICI: Nivolumab + TMZ + RT	mOS 28.9 months	CheckMate 548 [94]	No OS gain over TMZ + RT; MGMT-methylated
	Vaccines: DCVax-L	mOS 19.3 months	Phase III [92]	Phase III, with RT + TMZ; benefit in mOS, non-standard
	Vaccines: Rindopepimut + TMZ	mOS 20.0 months	ACT IV [92,106]	Phase III, no improvement in OS; not recommended for EGFRvIII+
	Cytokines: IFN-α + TMZ	mOS 26.7 months	Phase III [92]	Phase III, adjuvant after RT; benefit in mOS, non-standard
	Triple IT	No recurrence at 17 months	GIANT, ongoing [93]	Neoadjuvant to surgery; single case; ongoing trials

Legend: ACT: Adoptive cellular therapies; BEV: Bevacizumab; ICI: Immune checkpoint inhibitors; IT: Immunotherapy; OV: Oncolytic viruses; PFS-6: 6-month progression-free survival.

**Table 11 biomedicines-13-02136-t011:** Advanced imaging and theragnostic.

Modality	Application	Performance	Limitations
Multiparametric MRI	Diagnosis, recurrence, pseudoprogression	High sensitivity, RANO 2.0; DSC (90% sensitivity, 88% specificity), DWI (ADC >1200 × 10^−6^ mm^2^/s), MRS (low Cho/Cr and Cho/NAA) for pseudoprogression [47,107,108]	Cost, need for specialized interpretation
PET 18F-FET	Theranostics (general)	High specificity (~80–90%), PET RANO 1.0 [110,111]	Cost, accessibility
PET [11C]-MET	Diagnosis, recurrence/ pseudoprogression	~95% sensitivity/specificity for grading; high accuracy for recurrence [111,112]	Short half-life, accessibility
PET [18F]F-DOPA	Diagnosis, recurrence/ pseudoprogression	92% sensitivity, 75% specificity for recurrence [110]	Cost, need for additional studies
PET [18F]FACBC	Diagnosis, recurrence/ pseudoprogression	90% sensitivity, 83% specificity for recurrence [111]	Need for further studies, accessibility
PET [68Ga]-PSMA-11	Diagnosis, recurrence/ pseudoprogression	High uptake in high-grade gliomas [113]	Need for further studies
Multiparametric MRI/PET-guided RT	Personalized RT with dose escalation	mOS 23 months in a phase I trial using multiparametric MRI and [18F]-FDOPA PET; no OS benefit in phase III trials [48]	Cost, accessibility, need for phase III validation
Theranostics [131I]-IPA	Treatment, evaluation	mOS 16 months in rGBM [111,113]	Limited BBB penetration, need for validation
Theranostics (general)	Treatment, evaluation	Ongoing trials (e.g., [177Lu]-PSMA, [177Lu]-6A10, [177Lu]-NeoB) [112,113]	Limited BBB penetration, need for validation

Legend: ADC: Apparent diffusion coefficient; BBB: Blood–brain barrier; Cho: Choline; Cr: Creatine; DSC: Dynamic susceptibility contrast perfusion; DWI: Diffusion-weighted imaging; MRS: Magnetic resonance spectroscopy; NAA: N-acetyl aspartate; RANO 2.0: 2023 criteria for glioma evaluation; PET RANO 1.0: Criteria for amino acid PET.

## Data Availability

No new data were created or analyzed in this study. Data sharing is not applicable to this article as all data discussed are derived from publicly available sources cited in the references.

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
