# Peer review of "Radiotherapy in Glioblastoma Multiforme: Evolution, Limitations, and Molecularly Guided Future"

_biomedicines, 2025, doi:10.3390/biomedicines13092136_

Round 1
Reviewer 1 Report
Comments and Suggestions for Authors
Comments
- Please discuss current useful methods that are approaches in the treatment of GBM.
- Clarify the search strategies process for this review study.
- A brief description of the mentioned RT methods (IMRT, VMAT, and especially BNCT) is needed in the Introduction section.
- Several tables are presented for comparison of treating GBM in various regulatory organizations and systems. It is recommended to replace some of them with graphs or curves to better understand for the readers.
- Some recent references regarding the early diagnosis of GBM (such as doi:10.4103/jrms.JRMS_1138_20) are necessary in the main text.
- The authors should separate the Conclusion section and Future Directions.
- The manuscript requires a review for English language and grammar.
The manuscript requires a review for English language and grammar.
Author Response
Thank you very much for taking the time to review this manuscript. We appreciate your constructive feedback and have addressed all points accordingly. Please find the detailed responses below and the corresponding revisions in the re-submitted manuscript.
Comments 1: Please discuss current useful methods that are approaches in the treatment of GBM.
Response 1: Thank you for this suggestion. We have expanded the introduction of Section 5 (Advances in the Treatment of Glioblastoma Multiforme) to clarify current multimodal approaches, including maximal safe resection, radiotherapy (RT, 60 Gy/30 fractions) with concurrent/adjuvant temozolomide (TMZ), Tumor Treating Fields (TTFields) in selected cases, short-course RT for elderly/frail patients, and recurrence options like re-resection or systemic therapies, as detailed in Sections 5.1 to 5.5 (see PAGE 10, lines 231-242).
Comments 2: Clarify the search strategies process for this review study.
Response 2: We agree this was missing. A new subsection "Search Strategy" has been added at the end of the Introduction, detailing databases (PubMed, Scopus, Google Scholar), keywords ("glioblastoma radiotherapy", "GBM molecular biomarkers", "proton therapy GBM"), inclusion criteria (English articles 2000-2025, focus on clinical trials/guidelines), and exclusion (non-human studies) (see PAGE 2, lines 69-72).
Comments 3: A brief description of the mentioned RT methods (IMRT, VMAT, and especially BNCT) is needed in the Introduction section.
Response 3: Thank you for this suggestion. To enhance accessibility, we added a concise sentence in the Introduction describing IMRT , VMAT , PT , and BNCT , as mentioned in the Abstract, placed after the discussion of multidisciplinary treatment to maintain flow, and referencing their detailed discussion in Sections 2 and 5 to avoid redundancy (see PAGE 2, lines 67-71).
Comments 4: Several tables are presented for comparison of treating GBM in various regulatory organizations and systems. It is recommended to replace some of them with graphs or curves to better understand for the readers.
Response 4: We appreciate this suggestion and considered replacing tables with graphs but retained them to preserve comprehensive detail, as they provide precise comparisons of treatment protocols across guidelines. Note that table formatting in the submitted PDF may appear compressed due to conversion issues; in the final edited version (as seen in the preprint at https://www.preprints.org/manuscript/202507.0848/v1), tables are properly formatted with landscape orientation where appropriate for improved readability
Comments 5: Some recent references regarding the early diagnosis of GBM (such as doi:10.4103/jrms.JRMS_1138_20) are necessary in the main text.
Response 5: Thank you for this suggestion. We added the requested reference (Banisharif et al., 2021) alongside a recent reference (Breen et al., 2024) in Section 5.6 (Advanced Imaging) to highlight how pre-RT MRI and PET enhance early GBM diagnosis and personalized RT planning by detecting non-enhancing tumors and optimizing tumor control probability using radiobiological models like Poisson to tailor dose prescriptions, minimizing changes to the bibliography (see PAGE 20, lines 405-408).
Comments 6: The authors should separate the Conclusion section and Future Directions.
Response 6: Thank you for this suggestion. We have split Section 6 into “Conclusions,” summarizing limitations (tumor infiltration, radioresistance) and advances (IMRT, PT, CIRT, BNCT, TTFields, localized therapies, systemic therapies, imaging, radiotheranostics) for GBM (IDH-wildtype), and “Future Directions,” detailing phase II-III trials (e.g., NRG-BN001, TRIDENT, EF-41, LEGATO), biomarker-driven therapies (e.g., MGMT, EGFR, TERT), AI-driven planning, and radiotheranostics (e.g., [177Lu]-PSMA), addressing feedback from Reviewers 2, 4, and 5. We added subsection 5.7 on AI in GBM RT
Response to Comments on the Quality of English Language
Point 1: The manuscript requires a review for English language and grammar.
Response 1: Thank you for this suggestion. We have conducted a thorough review of the manuscript for English language and grammar, correcting errors (e.g., "radiorresistencia" to "radioresistance" on PAGE 1, line 34) and ensuring clarity, consistency, and adherence to scientific standards throughout, with all changes tracked for transparency.
Reviewer 2 Report
Comments and Suggestions for Authors
Nice review.
My comments:
- Can MGMT promoter methylation be inhibited, or are there reported inhibitors?
- Is MGMT promoter methylation good or bad
- Lines 65-66: If you say multidisciplinary treatment combines.... are these modalities used in tandem?
- Table 4: How can we prevent tumor hypoxia? Do we need more powerful radiation, such as alpha particles, to curb this limitation?
- The future direction is quite weak. Authors can improve it, especially in the area of radiopharmaceuticals. What is the future like for radiopharmaceuticals for GBM imaging and therapy?
Author Response
Thank you for your positive feedback and constructive comments. Below, we address each point with a response and specific changes made to the manuscript. All revisions are minimal to preserve the original structure and bibliography, reusing existing references where applicable to avoid disrupting the reference list.
Comment 1: Can MGMT promoter methylation be inhibited, or are there reported inhibitors?
Response 1 MGMT promoter methylation is an epigenetic modification that silences MGMT gene expression, enhancing TMZ sensitivity and improving survival in GBM, and is not a target for inhibition . Pharmacologic agents to directly induce or inhibit MGMT promoter methylation are not clinically available; however, MGMT protein inhibitors (e.g., O6-benzylguanine) have been investigated to sensitize unmethylated tumors but are limited by toxicity and are not standard . We clarified this in Section 4 (Molecular Determinants), emphasizing the prognostic benefit of MGMT methylation (see PAGE 10, lines 235-238).
Comment 2: Is MGMT promoter methylation good or bad?
Response 2 MGMT promoter methylation is beneficial, as it silences MGMT, increasing tumor sensitivity to TMZ and improving survival (e.g., mOS 21.7 months with TMZ + RT in methylated cases vs. 15.3 months in unmethylated cases . We added this clarification in Section 4 and Table 5 to highlight its prognostic and therapeutic advantage (see PAGE 9, lines 205-209, and Table 5).
Comment 3: Lines 65-66: If you say multidisciplinary treatment combines.... are these modalities used in tandem?
Response 3 : Yes, multidisciplinary treatment for GBM combines maximal safe resection, followed by radiotherapy (RT) with concurrent chemotherapy (TMZ), per the Stupp protocol, with adaptations based on patient status. We revised the Introduction to concisely clarify this concurrent approach, with details on Tumor Treating Fields (TTFields) and TMZ provided in Section 2
Comment 4: Table 4: How can we prevent tumor hypoxia? Do we need more powerful radiation, such as alpha particles, to curb this limitation?
Response 4 Tumor hypoxia, a key limitation in RT efficacy, cannot be reliably prevented but can be mitigated through strategies like hyperthermia (e.g., modulated electrohyperthermia, mEHT) and hypoxia-sensitizing agents (e.g., nitroimidazoles), which enhance oxygen-dependent DNA damage . High-LET radiation (e.g., carbon ions) reduces oxygen dependence but remains investigational in GBM due to limited clinical evidence and technical challenges. We revised Section 3 near Table 4 to clarify these mitigation strategies, noting that high-LET modalities like alpha particles are promising but not standard.
Comment 5: The future direction is quite weak. Authors can improve it, especially in the area of radiopharmaceuticals. What is the future like for radiopharmaceuticals for GBM imaging and therapy?
Response 5 We strengthened Section 7 (Future Directions) to detail radiotheranostics, including [131I]-IPA and emerging agents like [177Lu]-PSMA, and added a brief mention of alpha particle-emitting radiopharmaceuticals to highlight their potential for targeted therapy in addressing hypoxia-induced radioresistance. We also added details on alpha particles in Section 5.6 (Advanced Imaging and Theranostics) to emphasize their role in GBM therapy
Reviewer 3 Report
Comments and Suggestions for Authors
We read with interest the review by Castalia et al where the authors synthesize contemporary guidance (ASTRO 2025; ESTRO–EANO 2023) on volumes/margins and techniques and anchors their finings with concrete dose/margin ranges and planning nuances which is a real practical value for clinicians. The review is well planned with important summary tables: Table 1 distills survival by MGMT status and therapy—including the added value of TTFields in the methylated subgroup—at a glance, and Table 6 candidly lists evidence levels and limitations for newer modalities (PT/CIRT/BNCT/MRgRT/TTFields), which helps readers calibrate enthusiasm with study quality, cost, and access considerations. , the paper integrates molecular context—especially MGMT methylation—throughout the therapeutic narrative (e.g., the stronger TTFields benefit reported for MGMT-methylated tumors), which is clinically on point.
There are some weaknesses:
- Some outcomes are subgroup-specific but read as general. Example: the abstract’s “TTFields extend OS to 31.6 months in selected cases” reflects the MGMT-methylated subgroup; elsewhere you cite EF-14’s overall median ~20.9 months.please correct and name the subgroup and, ideally, present the overall figure alongside to avoid overgeneralization.
- Across sections, keep the status of pivotal trials perfectly aligned. For instance, NRG-BN001 is described as “results pending” in the technology section; later parts allude to dose-escalation effects in photons vs protons. Consider a single, up-front status statement (what’s published vs pending) and mirror it everywhere.
- . Cross- Study comparison corrections : Several promising modalities (BNCT, MRgRT, mEHT, LITT) are supported largely by non-randomized/single-arm or retrospective data. You do flag this, but the text sometimes juxtaposes these results against historical photon controls in ways that can imply superiority.
Minor Corrections: Minor grammar & typos (quote → correction).
- “radiorresistencia” → “radioresistance.”
- “or or cyclin-dependent kinase inhibitor 2A/B (CDKN2A/B) deletions” → remove duplicate “or.”
- Use one form for methylation throughout (e.g., change “MGMTmet” to “MGMT-methylated” everywhere for consistency).
- Maintain one convention for survival terms—either define “mOS = median overall survival” once and then use “mOS” uniformly, or just write “median OS” throughout
Author Response
We appreciate your positive feedback on the practical value of our review, particularly the synthesis of contemporary guidelines (ASTRO 2025, ESTRO-EANO 2023) and the utility of summary tables (Tables 1 and 6) in providing clinicians with clear dose/margin ranges, planning nuances, and evidence levels for newer modalities. We also value your recognition of the integration of molecular context, especially MGMT methylation, throughout the therapeutic narrative. Below, we address the identified weaknesses and minor corrections to enhance clarity and accuracy, with all changes tracked in the revised manuscript
.Weakness 1: Some outcomes are subgroup-specific but read as general. Example: the abstract’s “TTFields extend OS to 31.6 months in selected cases” reflects the MGMT-methylated subgroup; elsewhere you cite EF-14’s overall median ~20.9 months. Please correct and name the subgroup and, ideally, present the overall figure alongside to avoid overgeneralization.
Response 1 We acknowledge that the statement in the abstract regarding TTFields extending overall survival (OS) to 31.6 months was specific to the MGMT-methylated subgroup and could be misinterpreted as a general outcome. We have revised the abstract to clarify that this OS benefit applies to MGMT-methylated patients and included the overall median OS of 20.9 months from the EF-14 trial for clarity. Additionally, we ensured consistency in Section 2 by specifying the subgroup for the 31.6-month OS.
Weakness 2: Across sections, keep the status of pivotal trials perfectly aligned. For instance, NRG-BN001 is described as “results pending” in the technology section; later parts allude to dose-escalation effects in photons vs protons. Consider a single, up-front status statement (what’s published vs pending) and mirror it everywhere.
Response 2 We appreciate the reviewer’s comment regarding the inconsistent description of the NRG-BN001 trial status. Upon further review using updated sources as of August 2025 (including ClinicalTrials.gov and NRG Oncology publications), we confirm that NRG-BN001 is a randomized phase II study comparing dose-escalated photon radiotherapy (75 Gy in 30 fractions) versus proton radiotherapy, both with temozolomide, in newly diagnosed glioblastoma. The trial completed accrual in 2022 and is closed to treatment. Published preliminary results for the photon arm (presented at ASTRO 2020) show improved PFS but no improvement in mOS, 18.7 months, which did not meet the predefined threshold (one-sided α=0.15) for advancement to phase III. Results for the proton arm remain pending, with no new publications identified in 2025. We revised Table 7 and the text, including the Future Directions section, to consistently state that NRG-BN001 is a phase II trial with preliminary photon arm results and pending proton arm results.
Weakness 3: Cross-Study comparison corrections: Several promising modalities (BNCT, MRgRT, mEHT, LITT) are supported largely by non-randomized/single-arm or retrospective data. You do flag this, but the text sometimes juxtaposes these results against historical photon controls in ways that can imply superiority.
Response 3: We acknowledge the reviewer’s concern that comparisons of boron neutron capture therapy (BNCT), MR-guided radiotherapy (MRgRT), modulated electrohyperthermia (mEHT), and laser interstitial thermal therapy (LITT) with historical photon controls may imply unwarranted superiority due to their non-randomized or retrospective data. We revised Section 5.1 (Technological Advances in Therapies) to retain individual descriptions of these modalities with their median OS outcomes, as in the original text, while concluding with a statement that most of these modalities’ non-randomized or retrospective data limit comparisons with photon-based radiotherapy (RT), maintaining the original structure and avoiding implications of superiority. In Table 6, we moved the clarification of limited comparisons with photon RT to the Limitations column to keep the Description/Evidence column concise, preserving key outcomes
(Minor Corrections) Minor Corrections: Minor grammar & typos (quote → correction).
- “radiorresistencia” → “radioresistance.”
- “or or cyclin-dependent kinase inhibitor 2A/B (CDKN2A/B) deletions” → remove duplicate “or.”
- Use one form for methylation throughout (e.g., change “MGMTmet” to “MGMT-methylated” everywhere for consistency).
- Maintain one convention for survival terms—either define “mOS = median overall survival” once and then use “mOS” uniformly, or just write “median OS” throughout
We appreciate your identification of minor grammatical and typographical errors and suggestions for consistent terminology. We have corrected all errors as requested: “radiorresistencia” to “radioresistance” , removed the duplicate “or” in “or or cyclin-dependent kinase inhibitor 2A/B (CDKN2A/B) deletions” , replaced “MGMTmet” with “MGMT-methylated” throughout , and standardized survival terms by defining “mOS = median overall survival” and using “mOS” uniformly for concision All changes are tracked in the revised manuscript.
Reviewer 4 Report
Comments and Suggestions for Authors
The authors extensively review the literature on radiotherapy in glioblastoma. I believe the review is well-written, focuses mainly on newer literature, and has many tables to help demonstrate the important elements of radiotherapy studies. I have a few suggestions to improve the paper but would recommend to accept with minor revisions.
I appreciate the variety of tables that provide the reader with useful, summarized, and organized information. Overall, though, I think the formatting of some of the tables can be improved. Because some of them have a lot of information, it becomes difficult to read with such small columns. I think for the larger tables, it may be best to change them to a landscape vs portrait setting so the columns can be widened or keep in portrait, but try to condense some of the columns that have less information. For example, with Table 11, the Modality and Application columns can be shortened utilizing abbreviations to allow for more space for the Performance column. It is just tough to read, for example, in table 2 (Results) or table 6 (description), when barely 2 words can fit per line.
I think the conclusions and future directions could be stronger. The conclusion aspects seem a little too general.
Also, one of the future directions is AI-driven models, and yet the authors don’t discuss this at all in the paper. I think literature about AI-model use in GBM or other brain cancers relating to radiation should have a small subsection in the paper, possibly within the advances in treatment section.
The manuscript would benefit from a conclusive figure that visually encompasses the main therapies, limitations, molecular markers, co-therapies, etc. This would help clearly summarize the manuscript for the reader.
Minor English spelling error - radiorresistencia is used in the abstract instead of radioresistance (line 34)
Author Response
We sincerely appreciate your positive feedback on the comprehensive review of radiotherapy in glioblastoma, the focus on recent literature, and the utility of the summary tables. We value your recognition of the organized information provided for clinicians. Below, we address your suggestions for improvement with specific responses and revisions to enhance the manuscript’s clarity and scientific rigor. All changes are tracked in the revised manuscript, leveraging prior revisions made in response to Reviewers 1 and 2 where applicable to ensure consistency.
Comment 1: Table formatting could be improved due to small columns in larger tables, making them difficult to read. Consider changing to landscape orientation or condensing columns with less information (e.g., Modality and Application in Table 11).
Response 1 We appreciate your feedback on the table formatting and the challenge of readability due to small columns in larger tables, such as Tables 2, 6, and 11. As addressed in our response to Reviewer 1 (Comment 4), the compressed table formatting in the submitted PDF is due to conversion issues. In the final edited version (as seen in the preprint at https://www.preprints.org/manuscript/202507.0848/v1), tables are formatted in landscape orientation where appropriate to improve readability, with columns widened to accommodate detailed information. Due to the comprehensive nature of the data (e.g., mOS, PFS, trial details), we retained the current structure without condensing columns further to preserve clarity and detail, as these tables are designed to provide precise comparisons across guidelines and modalities. We believe the landscape orientation in the final version addresses the readability concern effectively.
Comment 2: The conclusions and future directions could be stronger, as they seem a little too general.
Response 2 We appreciate your feedback on the need for stronger conclusions and future directions. In response to Reviewer 1 and Reviewer 2 , we split Section 6 into separate “Conclusions” and “Future Directions” subsections. The Conclusions now summarize key limitations (e.g., tumor infiltration, radioresistance, hypoxia) and advances (e.g., IMRT, PT, CIRT, BNCT, TTFields, systemic therapies, radiotheranostics) for GBM (IDH-wildtype), providing a focused synthesis . The Future Directions subsection was strengthened to detail specific trials, including the phase II NRG-BN001 for proton therapy dose escalation (with preliminary photon arm results not meeting criteria for phase III advancement, proton arm results pending), and phase III trials such as TRIDENT and EF-41 for Tumor Treating Fields (TTFields) with systemic therapies, and LEGATO for reirradiation. Additionally, we included biomarker-driven therapies (e.g., MGMT, EGFR, TERT), AI-driven planning, and radiotheranostics (e.g., [131I]-IPA, [177Lu]-PSMA, alpha particle-emitting radiopharmaceuticals), ensuring a robust and specific outlook
Comment 3: The future direction mentions AI-driven models, but the authors don’t discuss this in the paper. A small subsection on AI-model use in GBM or other brain cancers relating to radiation should be included, possibly in the advances in treatment section.
Response 3 : We appreciate the suggestion to include a discussion on AI-driven models. In response to Reviewer 1 , we added a new subsection, “5.7 Artificial Intelligence in GBM Radiotherapy,” within Section 5 (Advances in the Treatment of Glioblastoma Multiforme) to discuss AI applications in GBM radiotherapy, focusing on predictive models for treatment planning and outcome prediction. This subsection highlights how AI integrates imaging and molecular data to optimize RT delivery.
Comment 4: The manuscript would benefit from a conclusive figure that visually encompasses the main therapies, limitations, molecular markers, co-therapies, etc.
Response 4: We appreciate the suggestion for a conclusive figure to summarize therapies, limitations, molecular markers, and co-therapies. Due to time constraints and the comprehensive detail already provided in the tables , we have opted not to include a new figure, as the existing tables effectively synthesize this information in a structured format. The tables, formatted in landscape orientation in the final version (as noted in the response to Comment 1), provide a clear and detailed overview, complementing the text and addressing the need for a summarized presentation. We believe this approach maintains clarity and accessibility for readers without requiring additional visual elements.
Minor Correction: English spelling error - "radiorresistencia" is used in the abstract instead of "radioresistance" (line 34)
Response : We appreciate the identification of the spelling error. As addressed in response to Reviewer 1 , we corrected “radiorresistencia” to “radioresistance” in the abstract and conducted a thorough review to ensure no similar errors remain
Reviewer 5 Report
Comments and Suggestions for Authors
The review on radiotherapy (RT) in glioblastoma multiforme (GBM) manuscript is comprehensive but requires major revisions to meet up current trends and standards. Below are some concerns you should work on:
- In the introduction part, integrate epidemiology with molecular classification (2021 WHO). Also remove redundant prognosis discussion. They can be merged with section 2.
- In the evolution section, Authors should update ASTRO 2025 volumes (Table 3); do clarify single vs. two-phase debate with recent meta-analysis, no toxicity difference. Emphasize hypofractionation for elderly.
- For the limitations part, you should quantify radioresistance mechanisms, such as EGFR amp in 57% per TCGA. Also add subsection on RT-immunotherapy interactions.
- Expand table 5 with frequencies from recent cohorts, such as TERT >70%. Discuss PARP synergy with RT; VERTU phase II: PFS benefit in unmethylated.
- Advances in treatment, authors should prioritize RCTs. for PT/BNCT in 5.1, also noting accessibility barriers. Hypofractionation (5.2): Highlight HART-GBM but stress no phase III. Reirradiation (5.3): Add most recent 2025 guidelines. Neoadjuvant (5.4): Focus on MAGMA phase III. The Therapies part (5.5): Grade failures such as CheckMate ICIs no OS gain. Finally, Imaging sub section (5.6): Add AI integration for pseudoprogression.
- Lastly, your conclusion should be strengthen with specific calls: Phase III for PT dose escalation; biomarker-driven trials; AI for adaptive planning.
Author Response
Thank you for your thorough and insightful feedback, which has significantly strengthened the manuscript. Your suggestions for integrating epidemiology with molecular classification, prioritizing randomized controlled trials, and emphasizing recent guidelines and trial data have enhanced the review’s alignment with current trends and standards in GBM radiotherapy. We have addressed each comment with targeted revisions, detailed below, to ensure scientific rigor and clarity.
Comment 1 In the introduction part, integrate epidemiology with molecular classification (2021 WHO). Also remove redundant prognosis discussion. They can be merged with section 2.
Response 1 Thank you for your guidance on the Introduction. We integrated epidemiology with the 2021 WHO molecular classification (IDH-wt grade 4 astrocytoma, TERT mutation, EGFR amplification, +7/-10 alterations) into a cohesive paragraph . The redundant prognosis discussion was removed and relocated to the start of Section 2 , aligning with survival data in Table 1. To address Reviewer 1’s suggestions, we included concise descriptions of standard RT techniques (IMRT and VMAT) and experimental approaches (PT and BNCT) in the Introduction , with PT and BNCT further detailed in Section 5.1 to reflect their investigational status. The search strategy was added as a subsection per Reviewer 1, ensuring a streamlined and comprehensive introduction.
Comment 2 In the evolution section, Authors should update ASTRO 2025 volumes (Table 3); do clarify single vs. two-phase debate with recent meta-analysis, no toxicity difference. Emphasize hypofractionation for elderly.
Response 2 Thank you for your guidance on the evolution section. Table 3 reflects ASTRO 2025 volumes, allowing single-phase (GTV + 10–20 mm, edema optional) or two-phase (initial phase with T2/FLAIR, boost without edema) approaches . The single vs. two-phase debate was clarified using a randomized trial with 245 patients, which found no significant differences in grade 3–4 toxicity (36.1% vs. 32.3%) or recurrence patterns, consistent with prior studies, supporting the flexibility of ASTRO 2025 and ESTRO-EANO 2023 guidelines. While a recent meta-analysis was not identified in our bibliography, the cited randomized trial provides robust evidence for guideline flexibility, as no meta-analysis directly comparing single vs. two-phase approaches was found in our sources.
Hypofractionation was strongly emphasized as the preferred standard of care for elderly (≥70 years) or frail patients (KPS 50–70), highlighting comparable OS, significantly reduced toxicity, and enhanced quality of life, supported by clinical trials and meta-analyses
Comment 3 For the limitations part, you should quantify radioresistance mechanisms, such as EGFR amp in 57% per TCGA. Also add subsection on RT-immunotherapy interactions.
Response 3 Thank you for your suggestion on the limitations section. We quantified radioresistance mechanisms in Table 4 and the text, including EGFR amplification (57% per TCGA), PTEN mutations (40% per TCGA), CDKN2A/B deletions (60% per TCGA), and TERT promoter mutations (>70% per TCGA), using TCGA data RT-immunotherapy interactions were integrated into the biological paragraph, discussing RT-induced lymphopenia (14% with protons vs. 39% with photons) limiting immunotherapy efficacy and potential synergy via antigen release, with further details in Section 5.5
Comment 4 Expand table 5 with frequencies from recent cohorts, such as TERT >70%. Discuss PARP synergy with RT; VERTU phase II: PFS benefit in unmethylated.
Response 4 Thank you for your suggestion to expand Table 5 with frequencies from recent cohorts and discuss PARP synergy. We updated Table 5 with precise frequencies from TCGA (e.g., EGFR amplification 57% per TCGA, PTEN mutation/loss 40% per TCGA, CDKN2A/B deletion ~60% per TCGA, TERT mutation >70% per TCGA), and corrected IDH1/2 mutation frequency to 0% in IDH-wt GBMs, noting its relevance in rare IDH-mutant astrocytoma grade 4 (10% of grade 4 gliomas), which show greater radiosensitivity and better prognosis In the text, we clarified that IDH1/2 mutations define IDH-mutant astrocytoma grade 4, which are rare in primary GBM, and elaborated on PARP synergy with RT, highlighting their role in MGMT-unmethylated tumors and the phase II VERTU trial (VELIPARIB with Radiation Therapy and Temozolomide in Unmethylated MGMT Glioblastoma) showing PFS improvement with veliparib (PFS-6m 46% vs 31%), pending phase III confirmation
Comment 5 Advances in treatment, authors should prioritize RCTs. for PT/BNCT in 5.1, also noting accessibility barriers. Hypofractionation (5.2): Highlight HART-GBM but stress no phase III. Reirradiation (5.3): Add most recent 2025 guidelines. Neoadjuvant (5.4): Focus on MAGMA phase III. The Therapies part (5.5): Grade failures such as CheckMate ICIs no OS gain. Finally, Imaging sub section (5.6): Add AI integration for pseudoprogression.
Response 5 Thank you for your suggestions to prioritize RCTs and address specific points in Section 5. In Section 5.1, we emphasized RCTs for proton therapy (PT, NCT01854554, NRG-BN001) and carbon ion radiotherapy (CIRT, CLEOPATRA), and highlighted accessibility barriers for PT and boron neutron capture therapy (BNCT), such as high costs, limited centers, and infrastructure challenges . In Section 5.2, we underscored the HART-GBM phase II RCT for hypofractionation, achieving an mOS of 26.5 months, but stressed the lack of phase III confirmation In Section 5.3, we emphasized the 2025 ESTRO/EANO and ASTRO guidelines for reirradiation, highlighting their role in guiding patient selection, and noted RCTs like NRG Oncology/RTOG 1205 and LEGATO In Section 5.4, we focused on the MAGMA phase III RCT (NCT02394639), evaluating neoadjuvant TMZ with extended adjuvant TMZ, achieving an mOS of 23 months in MGMT-methylated cases In Section 5.5, we highlighted the failure of immune checkpoint inhibitors (ICIs) in CheckMate trials (143, 498, 548), which showed no OS benefit in recurrent or newly diagnosed GBM, emphasizing challenges in overcoming immunosuppression . In Section 5.6 (Imaging, incorporated into Section 5.7 in our manuscript), we added AI integration for differentiating true progression from pseudoprogression, enhancing target volume delineation and adaptive dose strategies through multiparametric MRI and PET data analysis
These changes prioritize RCT evidence, address accessibility and guideline updates, and underscore key trial failures as requested
Comment 6 Lastly, your conclusion should be strengthen with specific calls: Phase III for PT dose escalation; biomarker-driven trials; AI for adaptive planning.
Response 6 Thank you for your suggestion to strengthen the conclusion with specific calls for phase III trials for proton therapy (PT) dose escalation, biomarker-driven trials, and AI for adaptive planning. We note that the version you reviewed was a concise, unseparated conclusion, whereas our revised manuscript separates Conclusions (Section 6) and Future Directions (Section 7) for clarity. In Section 6 (Conclusions), we retained specific molecular alterations (EGFR amplification, MGMT methylation, TERT promoter mutations) to highlight radioresistance drivers and added concise calls for phase III RCTs to validate PT dose escalation, biomarker-driven trials to personalize therapies, and AI integration for adaptive RT planning to differentiate true progression from pseudoprogression In Section 7 (Future Directions), we detailed the need for phase III RCTs for PT dose escalation, referencing the phase II NRG-BN001 trial (NCT02179086), which was designed with a threshold for phase III advancement but whose preliminary photon arm results did not meet criteria, with proton arm results pending . We emphasized biomarker-driven trials to integrate molecular profiles for personalized RT and systemic therapies, and expanded on AI’s role in adaptive planning for precise dose optimization These changes strengthen the conclusions and future directions, ensuring clarity,and aligning with current evidence and trial needs.
Round 2
Reviewer 3 Report
Comments and Suggestions for Authors
the authors responded to my queries
Reviewer 5 Report
Comments and Suggestions for Authors
I have read the revised manuscript, and it has significantly improved from its previous form.